# Hydrogen-bonds mediate liquid-liquid phase separation of mussel derived adhesive peptides

Qi Guo[1,5], Guijin Zou [2,5], Xuliang Qian [3], Shujun Chen[1], Huajian Gao [2,3] ✉ & Jing Yu [1,4] ✉

Marine mussels achieve strong underwater adhesion by depositing mussel foot proteins (Mfps) that form coacervates during the protein secretion. However, the molecular mechanisms that govern the phase separation behaviors of the Mfps are still not fully understood. Here, we report that GK-16*, a peptide derived from the primary adhesive protein Mfp-5, forms coacervate in seawater conditions. Molecular dynamics simulations combined with point mutation experiments demonstrate that Dopa- and Gly- mediated hydrogen-bonding interactions are essential in the coacervation process. The properties of GK-16* coacervates could be controlled by tuning the strength of the electrostatic and Dopa-mediated hydrogen bond interactions via controlling the pH and salt concentration of the solution. The GK-16* coacervate undergoes a pH induced liquid-to-gel transition, which can be utilized for the underwater delivery and curing of the adhesives. Our study provides useful molecular design principles for the development of mussel-inspired peptidyl coacervate adhesives with tunable properties.

Liquid–liquid phase separation (LLPS) known as coacervation is involved in the biogenesis of many extracellular materials, including various silks[1–3], wings of butterfly[4], squid beak[5], and marine adhesives[6–8]. Marine mussels make their adhesive byssus by secreting a family of mussel foot proteins (Mfps) with a microfluidic-like fabrication process using an organ called the mussel foot[9,10]. Mfps are rich in a post-translationally modified amino acid L−3,4-dihydroxyphenylalanine (Dopa), which plays an essential role in the mechanical properties of the plaque[11]. In the mussel foot, Mfps form coacervates in the internal environment of the foot distal depression during secretion. After spatial-temporally regulated[10] secretion of Mfps, the seawater triggers the solidification of the secreted coacervates and further plaque maturation[11]. However, the molecular mechanisms that govern the LLPS properties of Mfps are not well understood[11,12].

Dopa plays the essential role in the adhesive and cohesive properties of Mfps via various interactions including hydrogen bonds, π-π interactions, π-cation interactions, and metal coordination[11,13–17]. The strong wet adhesion of Mfps has inspired numerous studies on making Dopa or catechol-containing synthetic adhesives[18–20]. In addition, Dopa is critical to the LLPS behavior of many Mfps. Dopa-mediated bidentate hydrogen bonding regulates the LLPS of Pvfp-5β, an primer adhesive protein from Asian green mussel Perna viridis[7]; Recombinant Mfp-1 from Californian mussel Mytilus californianus forms self-coacervate via cation-π interaction between Dopa and Lys residues[21]. Mfp-3slow, a matrix protein of the adhesive plaque, can also form coacervate driven by hydrophobic interactions[8].

Mfp-5 is the first secreted protein to initiate the wet adhesion of mussels during the plaque formation[22,23]. It has the highest Dopa

[1]School of Materials Science and Engineering, Nanyang Technological University (NTU), 50 Nanyang Avenue, Singapore 637553, Singapore. [2]Institute of High Performance Computing, A*STAR, Singapore 138632, Singapore. [3]School of Mechanical and Aerospace Engineering, Nanyang Technological University (NTU), 50 Nanyang Avenue, Singapore 639798, Singapore. [4]Institute for Digital Molecular Analytics and Science, Nanyang Technological University (NTU), 50 Nanyang Avenue, Singapore 637553, Singapore. [5]These authors contributed equally: Qi Guo, Guijin Zou. ✉e-mail: huajian.gao@ntu.edu.sg; yujing@ntu.edu.sg

content (~30%) and the highest adhesion among all the Mfps[24]. However, the LLPS of Mfp-5 has never been clearly identified. Here we report that a short peptide with the important adhesive motif from Mfp-5, denoted GK-16* (GY*KGKY*Y*GKGKKY*Y*K, Y* represents Dopa), can form coacervate in seawater conditions. Molecular dynamics (MD) simulations show that the LLPS of GK-16* is mainly driven by Dopa-mediated bidentate H-bonds. The LLPS behavior of GK-16 can be systematically tuned by point mutation of the Dopa residues. Due to a high Lys content, GK-16* coacervates show pH-dependent properties, undergoing a liquid-to-gel transition when raising the solution pH from 3 to seawater pH. The sequence- and pH-dependence of GK-16* not only elucidate the molecular mechanisms of the LLPS behavior of mussel adhesive proteins, but can also be utilized for the design and delivery of next-generation underwater adhesives.

## Results and discussion

### Liquid–liquid phase separation of GK-16* peptide

Mfp-5 is the first secreted protein to initiate the mussel adhesion (Fig. 1a)[23]. As an interfacial adhesive primer, Mfp-5 locates at the interface between mussel byssus and target surfaces, taking a major role in achieving robust interfacial adhesion via Dopa-mediated adhesive interactions between the plaque and the target surface. According to the sequence and amino acid composition of Mfp-5 (30 mol% of Dopa, 21 mol% of Gly and 20 mol% of Lys)[24], we designed a hexadecapeptide named GK-16 (GYKGKYYGKGKKYYYK), which is highly representative of the sequence of Mfp-5 (Fig. 1b and Supplementary Fig. 1). The Tyr residues were converted into Dopa (Y*) by mushroom tyrosinase enzymatic treatment[25]. The Dopa-converted peptide is denoted as GK-16*. We tested the adhesive and phase separation behaviors of GK-16* at pH 3 (Fig. 1c) as natural Mfps are stored and secreted in acidic environment with pH as low as 3[23]. Without additional salt, GK-16* was soluble in pH 3 solution. However, by adding 600 mM NaNO3 into the solution to mimic the seawater salinity, we observed coacervate microdroplets in the solution under optical microscope (Fig. 2a). LLPS occurred when GK-16* concentration was beyond 0.5 mg/mL and NaNO3 or NaCl concentration was above 100 mM (Supplementary Fig. 2). With the increasing peptide concentration, the threshold salt concentration of phase boundary shows a downtrend (Fig. 2b). Circular dichroism (CD) and Fourier-transform infrared spectroscopy (FTIR) spectra reveal that the second

structure of the peptide changes from mainly random-coil to a mixture of coil and β-sheet structures as the ionic strength of the solution increased from 1 mM to 600 mM (Fig. 2c, Supplementary Fig. 3, and Supplementary Table 1). At low ionic strength, GK-16 and GK-16* mainly contain random-coil structures due to the electrostatic repulsion between the lysine residues (Supplementary Fig. 4), which is effectively screened by the added salt and therefore promotes a change of secondary structures. Zeta potential measurements show that the electrokinetic potential decreased with the increasing salt concentration, confirming the screening effect of salts (Supplementary Fig. 6). The coacervation of GK-16* is mainly driven by Dopa-mediated interactions, as unmodified GK-16 did not show any LLPS behavior in pH 3 solution at any peptide and salt concentration investigated (Supplementary Fig. 7).

We performed surface forces apparatus (SFA) experiments to investigate the mechanical properties of the GK-16* coacervates (Fig. 2d). SFA force-distance profiles show that in pH 3 HCl solution with no additional salt, GK-16* formed a peptide monolayer on mica surface, with a 2 nm film thickness and a small adhesion force of ~1 mN/m up on separating two surfaces. The adhesion of GK-16* on mica surface is due to the Dopa and positive lysine residues. After increasing ionic strength of the solution to 600 mM by injecting a calculated amount of 5 M NaNO3 solution into the gap solution between two surfaces, GK-16* formed coacervate, and the film thickness increased to ~50 nm. Upon separation, a viscous jump-out was observed with a normalized adhesion force (F/R) of 1.3 mN/m, indicative of the capillary adhesion caused by the GK-16* coacervate. The low adhesion force shows that the GK-16* coacervate has low interfacial energy ($\gamma = 0.10$ mJ/m²) in solution, which can facilitate the spreading of the coacervate adhesives on different substrates[8,26]. Further increasing the ionic strength to 2000 mM leads to a longer-ranged repulsion between two surfaces with abolished adhesion force, indicating that the GK-16* coacervate becomes more gel-like at higher salt concentrations. This liquid-to-gel transition is likely due to the strong screening effect of high salt content in solution which reduces the electrostatic repulsion between Lys residues and promotes the further association of GK-16* chains.

To further test the hypothesis that reduced electrostatic interactions between GK-16* chains promote the LLPS of the peptide, we explored the phase separation behavior of GK-16* at pH 7.4 using phosphate buffer. Zeta potential measurements validated that the peptide carried less positive charge at pH 7.4 than at pH 3 (Supplementary Fig. 6). The phase separation regime of GK-16* was expanded at pH 7.4 (Fig. 2b). However, instead of forming spherical coacervate droplets, GK-16* formed gel-like precipitates in the phase separation regime (Supplementary Fig. 8). We initially assumed Dopa would be oxidized into Dopaquinone at pH 7.4. However, ultraviolet–visible (UV–Vis) spectra (Supplementary Fig. 9) showed that the Dopa peak at 280 nm remained unchanged, and there was no Dopaquinone peak occurring at 400 nm. The increased stability of Dopa in the GK-16* coacervate agrees well with recent discovery that the inner chemical environment of coacervate could shield Dopa from oxidation[27].

As unmodified GK-16 cannot form LLPS, we speculated that the Dopa residues can form bidentate hydrogen bonds and function as the "sticker" in the LLPS process. π-cation interaction is unlikely to play essential roles in the LLPS process of GK-16* given the fact that the π-cation interaction between Tyr and Lys is stronger than that of Dopa[14]. To tune the Dopa-mediated hydrogen bonds, we used urea to disrupt the hydrogen bonds between GK-16* chains. To mimic the seawater environment, we chose the ionic strength of 600 mM to ensure the appearance of GK-16* coacervate microdroplets. Adding 200 mM urea to the solution resulted in more fluid-like droplets and promoted droplets coalescence (Fig. 2a). At sufficiently high concentrations, urea could lead to the dissolution of the coacervate droplets (Fig. 2e), albeit the critical urea concentration depended on the concentration of GK-

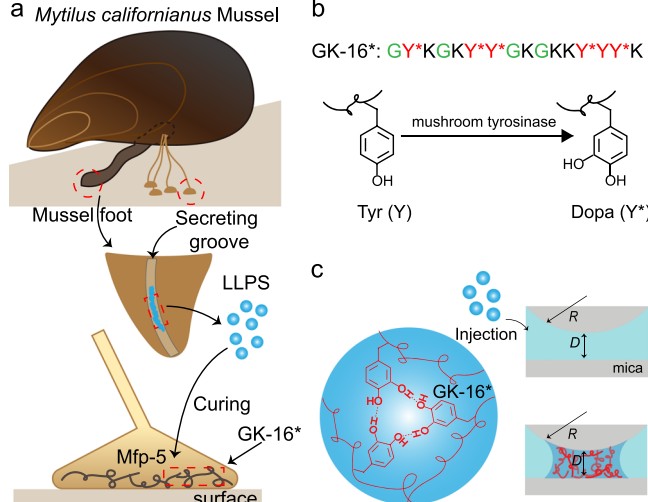

**Fig. 1 | GK-16* peptides derived from Mfp-5, a primary adhesive protein in the mussel plaque. a** *Mytilus Californianus* mussels secret and transport phase separated Mfps through mussel foot groove. **b** The sequence of GK-16* peptide and the enzymatic modification process of converting Tyr to Dopa. **c** GK-16* can form coacervate in seawater conditions.

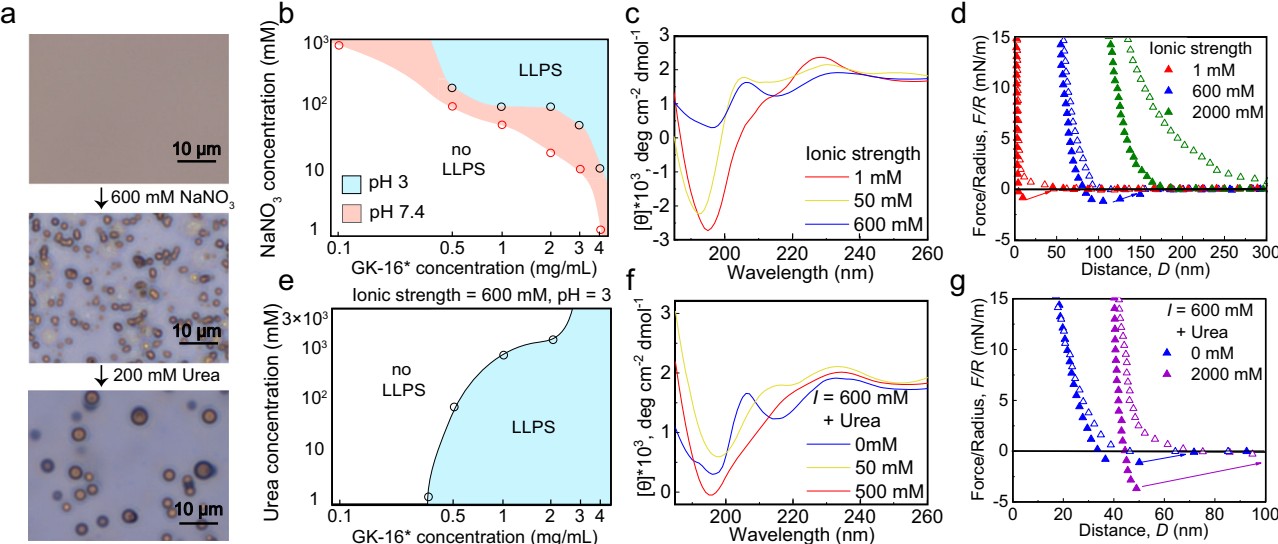

**Fig. 2 | LLPS of GK-16*. a** By adding salt and urea, GK-16* forms coacervates with tunable morphologies. Three experiments were repeated independently with similar results. **b** Phase diagram of GK-16* with different peptide and salt concentrations at pH 3 and 7.4. **c** CD spectra of GK-16* (1 mg/mL) in solutions with different ionic strengths. **d** SFA force-distance profiles of GK-16* (2 mg/mL) in solutions with different ionic strengths. **e** Phase diagram of GK-16* at different peptide and urea concentrations with fixed ionic strength (600 mM) and pH (pH = 3). **f** CD spectra of GK-16* (1 mg/mL) in solutions with fixed ionic strength (600 mM) and different urea concentrations. **g** SFA measurements of GK-16* (2 mg/mL) in solutions with 0 and 2000 mM urea.

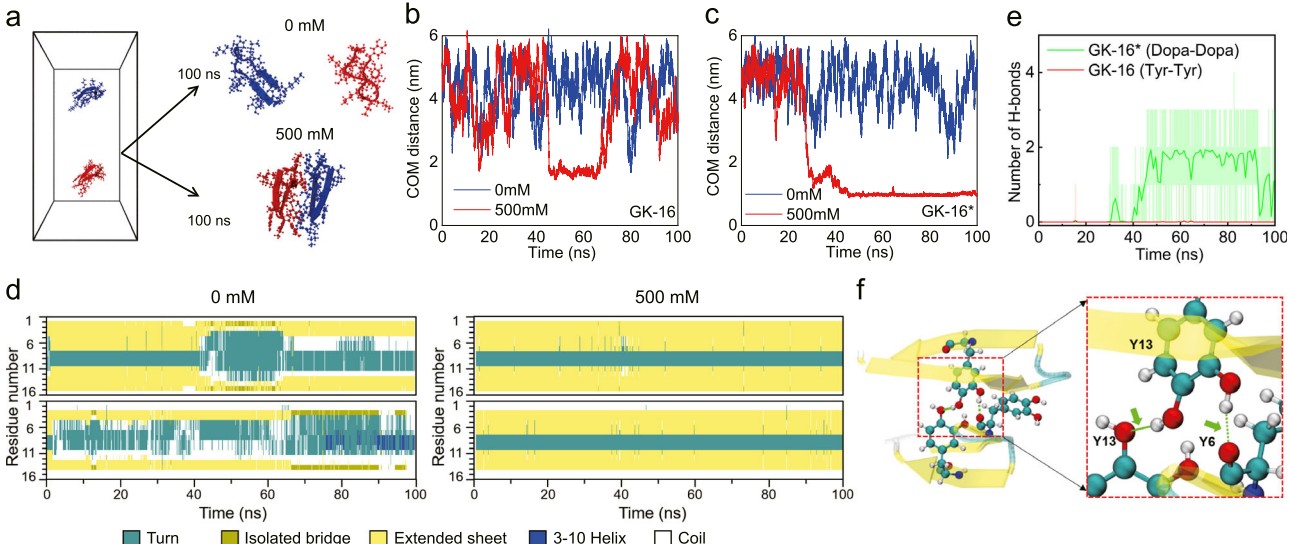

**Fig. 3 | MD simulations of the pair interactions of GK-16 and GK-16* peptides. a** Simulation models; left: simulation box with initially separated peptides; right: the final snapshots of GK-16* under 0 mM or 500 mM ionic strength at 100 ns. Water molecules and ions are not shown for clarity. The representative temporal evolutions of COM distances between the pairs of peptides with 0 mM or 500 mM ionic strength, for GK-16 (**b**) and GK-16* (**c**). **d** Residue maps of the secondary structures as a function of time for GK-16* under 0 mM and 500 mM ionic strength in simulations. **e** The temporal evolutions of the number of H-bonds formed between Dopa residues in GK-16* and Tyr residues in GK-16 (transparent line) and the corresponding average over 1 ns (solid line) during the interaction process. **f** Representative configurations of the associated GK-16*; the right panel shows a magnified view of the bidentate H-bonds, green arrows indicating the H-bonds. Atoms in Dopa residues are colored as follows: hydrogen, white; oxygen, red; carbon, cyan; nitrogen, blue.

16*. CD results reveal that by adding urea from 0 mM to 500 mM, the secondary structures of GK-16* changed from mainly β-sheets to random-coil structures (Fig. 2f), resulting in dissolution of the coacervate droplets. SFA (Fig. 2g) showed with the addition of 2000 mM urea, the film thickness of the coacervate increased from ~20 nm to ~40 nm, with slightly increased adhesion upon separation. The increasing of film thickness in the SFA experiment also suggests the coalescence of GK-16* microdroplets, which agrees well the growth of droplet size observed under an optical microscope.

## MD simulations of peptide association

We performed molecular dynamics (MD) simulations to investigate the governing molecular interaction mechanisms behind the observed LLPS behaviors of GK-16*. As demonstrated in the studies of polyelectrolyte complex[28,29], the interaction and association behaviors between the pairs of biomolecules are closely related to their phase separation[17]. We performed a series of MD simulations consisting of pairs of peptides (Fig. 3a and Supplementary Fig. 11); the details of the model are supplied in the Methods. At low ionic strength (0 mM), for

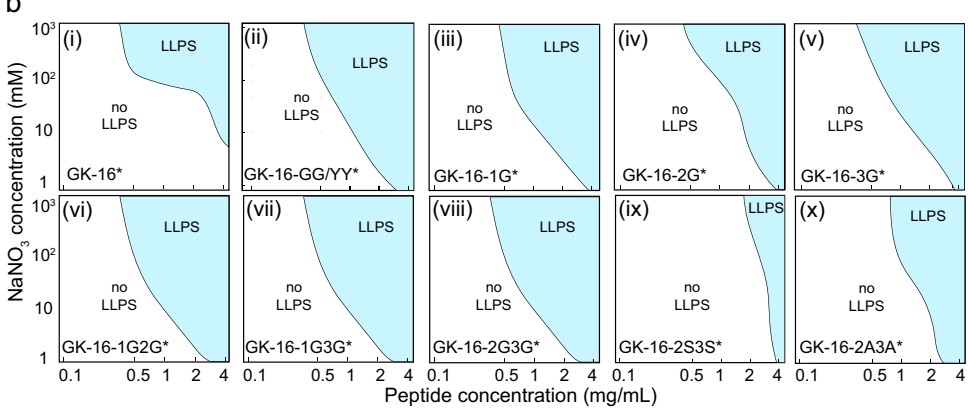

Fig. 4 | LLPS behaviors of mutated GK-16* variants. a The list of GK-16* variants with modified sequences, and corresponding LLPS and association behaviors in experiments and MD simulations. Lines with * and # represents 900 and 300 mM ionic strength respectively, the others were under 500 mM ionic strength to capture the difference. b Phase diagrams of GK-16* variants: (i) GK-16*; (ii) GK-16-GG/YY*; (iii) GK-16-1G*; (iv) GK-16-2G*; (v) GK-16-3G*; (vi) GK-16-1G2G*; (vii) GK-16-1G3G*; (viii) GK-16-2G3G*; (ix) GK-16-2S3S*; (x) GK-16-2A3A*.

either GK-16 or GK-16* peptides, the electrostatic repulsion due to the positively charged Lys residues dominate the interaction processes. The peptides fluctuated randomly in the simulation box, with no stable associated states as shown in the time evolutions of the center of mass (COM) distances of the peptides (Fig. 3b, c, Supplementary Fig. 11a, c). When we increased the ionic strength to 500 mM in simulations, the charge screening reduces the effect of electrostatic repulsion, allowing the peptides to bind, decreasing COM distances to about 1.7 nm or 0.97 nm for GK-16 or GK-16*, respectively. However, the association of GK-16 is transient, less compact and can easily be disassociated by thermal fluctuations (Fig. 3b and Supplementary Fig. 11b). In contrast, the pair of GK-16* can bind persistently, maintaining a stable complex by stacking two parallel beta-sheet structures throughout the duration of the simulations (Fig. 3a, c, Supplementary Fig. 11d), which is consistent with the experimental observation that the Dopa-modification of GK-16 can lead to phase separation (Fig. 2b). Additionally, the time evolutions of the secondary structures of GK-16* indicate that high ionic strength is necessary to maintain the beta-sheet structures of GK-16* (Fig. 3d), consistent with the CD spectra experiments (Fig. 2c). We further calculated the H-bonds formed between the Dopa or Tyr residues from the two peptide chains. In contrast to Tyr residues in GK-16 with almost no H-bonds formed (Fig. 3e), the Dopa residues in GK-16*,

due to the two hydroxyl groups, can form bidentate H-bonds (Fig. 3e, f), leading to the stable association of GK-16* peptides and taking function as the "stickers". In summary, these MD simulations showed that Dopa residues play essential roles in the association of GK-16* peptides through the formation of bidentate H-bonds, which correlates with the phase separation abilities of the peptides.

**Point mutation of GK-16* variants**
Both experiments and simulations reveal that the Dopa residues play critical roles in the LLPS of GK-16*. By replacing two Gly residues of GK-16* into Dopa, GK-16-GG/YY* showed a slightly larger LLPS region (Fig. 4b) as increasing the amount of Dopa residues could promote the hydrogen bonds. Completely replacing all the Tyr resides by Gly in the GK-16 sequence resulted in no LLPS in any condition tested, which agreed well with the observation that unmodified GK-16 did not phase-separate in solution (Fig. 4a). All the Dopa/tyrosine residues locate in three locations in the GK-16* sequence, denoted as location-1, -2, and -3, which contain 1, 2, and 3 Dopa residues, respectively. We performed systematic point-mutation tests on GK-16* by replacing the Dopa residues with Gly, which is commonly regarded as a "spacer" in polypeptides[30]. Mutating the Dopa in one of the three positions did not significantly alter the LLPS behaviors of the three GK-16* variants,

denoted as GK-16-1G*, GK-16-2G*, and GK-16-3G*. All three variants showed LLPS upon addition of salt at pH 3. However, the LLPS region of the three variants expanded toward lower salt and peptide concentrations (Fig. 4b). More surprisingly, GK-16-2G can form coacervates exhibiting slightly reduced LLPS region in the phase diagram even without the Dopa conversion (Supplementary Fig. 13), likely due to the enhanced backbone flexibility by enriching Gly at the center of the peptide. The enhanced LLPS behavior of the Dopa-replaced GK-16 variants indicates that Gly and Dopa may act synergistically in the association of Dopa-containing peptides.

To further test the synergy between Dopa and Gly, we replaced the Dopa residues by Gly in any two of the three Dopa-containing positions (GK-16-1G2G*, GK-16-1G3G*, and GK-16-2G3G*). Phase separation experiments showed that all the three peptides exhibited enhanced LLPS behavior with LLPS regimes further expanding toward lower peptide concentrations. In addition, we replaced the Gly in GK-16-2G3G* with less flexible Ala or Ser residues[30,31]. With such mutations, GK-16-2S3S* showed a much narrower LLPS region than GK-16-2G3G* (Fig. 4b), while GK-16-2A3A* showed a larger phase separation region than GK-16-2S3S*, but a smaller LLPS region than GK-16-2G3G* (Fig. 4b). The differences in the phase separation regimes of GK-16-2S3S* and GK-16-2A3A* is likely due to the stronger hydrophobicity of Ala. Furthermore, replacing all the Gly residues to Ser in GK-16* (SK-16*) completely eliminated the LLPS of the peptide, demonstrating the importance of glycine in the phase separation of GK-16*. Glycine-rich polypeptides have high backbone flexibility and can easily adopt helical or β-sheet structures[32,33], which promotes self-assembly of the peptides. This assembly could be further stabilized via Dopa-mediated bidentate H-bonds (Figs. 2c and 3f). CD spectra on the Dopa-replaced GK-16 variants confirm that the Dopa to glycine replacement could promote the formation of the localized β-sheet or β-turn structures (Supplementary Fig. 14f, g).

We also explored the effect of Lys in the LLPS behavior of GK-16*. Completely replacing the positively charged Lys to Gly rendered the peptide insoluble in aqueous solution. We then replaced 2 Lys residues of GK-16* with Gly (GK-16-KK/GG*) to reduce the charge of the peptide. GK-16-KK/GG* showed a slightly larger phase separation region than that of GK-16* because of the reduced electrostatic repulsion (Supplementary Fig. 15), which is also supported by the association of GK-16-KK/GG* under lower ionic strength in simulations (Fig. 4a, Supplementary Figs. 19 and 21k). Such point mutations demonstrate the importance of positive charges in balancing the solubility and phase separation behavior of the peptide.

## MD simulations of GK-16* variants

Since directly modeling the process of LLPS is beyond the computational capability of all-atom MD simulations[34–36], we focused on the study of the structures of GK-16 variants and their Dopa-modified derivatives as well as their interaction and the association behaviors, which are foundations for the more complicated, higher-order assembly in the phase separation. The combined analysis of the root-mean-square deviation (RMSD), radius of gyration and root-mean-square fluctuation (RMSF) of the peptides in MD simulations revealed that the Dopa-modification does not alter the peptide flexibility significantly (Supplementary Figs. 16 and 17), but that replacing the Tyr/Dopa residues by Gly in the sequence increases the flexibility of peptides (Supplementary Fig. 18), consistent with the observation in previous studies[30–33,37]. Among the three positions, position-2 at the center is most important for the flexibility as GK-16-2G exhibits more substantial fluctuations compared to GK-16-1G and GK-16-3G (Supplementary Fig. 18). In addition, replacing the Gly with Ser or Ala in general resulted in less peptide fluctuations due to the hydroxyl or methyl side group (Supplementary Fig. 18).

We further conducted a series of pair-interaction simulations of the original peptides and the Dopa-modified derivatives under the

same conditions, which enable us to identify the roles of each residue in the inter-peptide association. As demonstrated in Fig. 4a and Supplementary Figs. 19–21, we found that the association of the peptides closely correlates with the phase separation in experiments. Since Dopa-modification mostly changes the inter-peptide interactions by varying the H-bond formations via the extra hydroxyl group, we did further analysis on the number of H-bonds formed between each residue (Fig. 5 and Supplementary Figs. 22 and 23). For GK-16*, consistent with previous results, H-bonds are mostly formed between the Dopa residues (Figs. 3e and 5a). The number of H-bonds formed between the side chain and the backbone is similar to that between the side chains (Fig. 5a(i), b(i), Supplementary Fig. 23), suggesting that the two hydroxyl groups of Dopa residue can form H-bonds with both the side chain and the backbone. This extra hydrogen bond capacity enables Dopa to be a much stronger "sticker" than Tyr. With increased content of Dopa, more hydrogen bonds were observed between Dopa and other residues (Fig. 5a(ii) and Supplementary Fig. 23). The combined experimental and simulation results of GK-16-G showed that Dopa stickers are indispensable for the phase separation (Fig. 4a).

Interestingly, both GK-16-2G and GK-16-2G* can associate in MD simulations and LLPS in experiments (Fig. 4a, Supplementary Figs. 13 and 19–21), which suggests that Gly residues, although, commonly regarded as "spacers"[30], can facilitate sticker-sticker interactions, making the weaker sticker Tyr strong enough to associate. We speculated that this surprising synergistic effect is likely a result of the substantial flexibility of the peptide induced by the Gly mutation at the center (Supplementary Fig. 18). This speculation is supported by the discovery that the H-bonds stabilizing the association mainly come from the peptide backbone (Fig. 5a(iv), b(iii), Supplementary Figs. 22a and 23). When Dopa residues at position-1 or 3 are replaced by Gly (GK-16-1G*, GK-16-3G*), the H-bonds formed between the peptides mostly come from Gly and Dopa residues (Fig. 5a(iii) and (v), b(ii)). As long as the peptide has two positions being replaced by Gly (GK-16-1G2G*, GK-16-1G3G* and GK-16-2G3G*), the H-bonds related to the side chain of Dopa residues are negligible, as expected (Fig. 5a(vi–vii), Supplementary Fig. 23). Consistent with the case of GK-16-2G, due to the high backbone flexibility arising from the high content of Gly residues, the H-bonds that drive the peptides association mainly come from the backbone (Fig. 5a, Supplementary Fig. 23). To test the effect of Gly, we further mutated Gly by another common spacer Ser at positions-2 and −3 (GK-16-2S3S*). There are two expected competing roles for the extra hydroxyl side group of Ser residue: on one side, it increases the hydrogen bond capacity while on the other side, it also makes the peptide less conformable (Supplementary Fig. 18)[30]. Our simulations suggest that the latter effect will be more important as GK-16-2S3S* exhibits a relative loose association compared to GK-16-2G3G* (Supplementary Figs. 19 and 21), with slightly fewer H-bonds and most of the H-bonds formed between the side chain and backbone (Fig. 5a (viii), ix), 5b (iv) and Supplementary Fig. 23). Completely changing Gly to Ser leads to no association in simulation or LLPS in experiments, further confirming that the role of Gly is more than just a spacer. Mutating Gly by Ala at position-2 and −3 (GK-16-2A3A*) will help the backbone hydrogen bond formations and association (Fig. 5a(x), b(v) and Supplementary Figs. 19, 21i, 23) likely due to the hydrophobic interactions. These simulation results, combined with our experimental observations, reveal the interaction modes between pairs of various mutated GK-16* peptides and demonstrate the key role of Gly and Dopa residues in determining the peptide association as well as the related phase separation behaviors. Given the high glycine and Dopa contents in many of the Mfps[11], such synergy may also contribute to the phase separation of natural Mfps.

## GK-16* based pH-responsive coacervate adhesives

The liquid-to-gel transition of GK-16* coacervate upon increasing the solution pH can facilitate the delivery of the coacervate adhesives: the

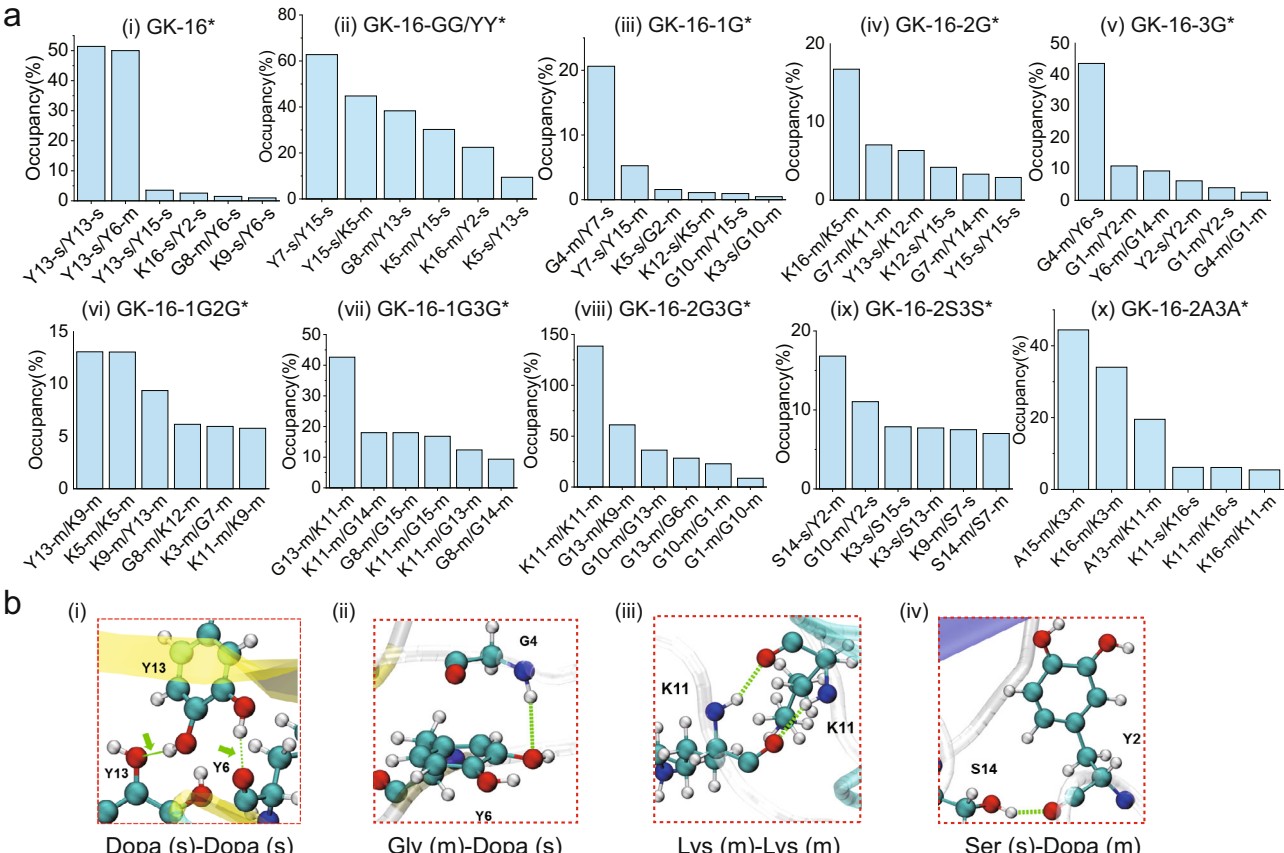

**Fig. 5 | H-bond formations between pairs of GK-16\* and its variants in MD simulations. a** The representative occupancy of most probable H-bond between different residues was obtained from the last 40 ns of simulations (the associated pairs) of (i) GK-16\*; (ii) GK-16-GG/YY\*; (iii) GK-16-1G\*; (iv) GK-16-2G\*; (v) GK-16-3G\*; (vi) GK-16-1G2G\*; (vii) GK-16-1G3G\*; (viii) GK-16-2G3G\*; (ix) GK-16-2S3S\*; (x) GK-16-2A3A\*. The occupancy representing the overall H-bond probability through simulations (see "Methods"). **b** Representative simulated snapshots of the most probable H-bonds between the corresponding residues (m signifies main chain, and s side chain). (i) H-bonds formed among Dopa side chains; (ii) H-bonds between Gly main chain and Dopa side chain; (iii) H-bonds between Lys main chains; (iv) H-bonds between Ser side chain and Dopa main chain.

adhesives can be delivered underwater at low pH; and upon delivery, the solvent exchange between the coacervate adhesives and the environment solution can increase the pH of the coacervate, which can induce the liquid-to-gel transition of the coacervate adhesives and therefore cure the adhesives on surfaces. To demonstrate this potential, we prepared a GK-16\* coacervate in pH 3 solution with 200 mM urea. At pH 3, the coacervate exhibited liquid-like property with reproducible viscous jump-outs and adhesion force of 1.64 mN/m in the SFA measurements (Fig. 6a and Supplementary Fig. 24a). We then switched the solution to phosphate-buffered saline (PBS) buffer (100 mM, pH = 7.4) while keeping two surfaces in contact. After equilibrium for 30 min, the solvent exchange between the coacevate phase and the surrounding solution would induce the liquid-to-gel transition of the GK-16\* coacervate. Upon separating two surfaces in PBS buffer, a nearly fifteen-fold increase of the adhesion force reaching ~ 26 mN/m was measured (Fig. 6a). Such strong adhesion was only measured in the first separation, and the adhesion forces measured in the subsequent force runs rapidly decreased to zero (Supplementary Figs. 24b and 25). This is due to the cohesive failure of the gel-like GK-16\* coacervate adhesive at pH 7.4 during the first separation of two surfaces. The gel-like coacervate adhesive could not coalesce upon bringing two surfaces into contact, leading to the abolished adhesion force. It is worth mentioning that such pH-responsive adhesiveness would require balanced molecular interactions between the peptides. Too strong interactions would lead to gel-like coacervates at high salt concentrations even at pH 3, which abolished the adhesion of the peptide coacervates (Supplementary Table 3, Supplementary Fig. 26).

Apart from GK-16\*, GK-16-GG/YY\* and GK-16-2G\* also demonstrated such pH-responsive liquid-to-gel transition (Supplementary Fig. 27).

The SFA measurements demonstrate the potential of using GK-16\* as a pH-responsive coacervate adhesive (Fig. 6b). Stored under pH 3, GK-16\* would form liquid-like coacervates via a delicate balance between Dopa-mediated hydrogen bonds and electrostatic repulsion between the lysine residues. While applying the GK-16\* coacervate adhesive into aqueous solutions with physiological or neutral pH, the coacervate adhesive can achieve strong adhesion on surfaces via Dopa-mediated bidentate bindings. The liquid-to-gel transition of the GK-16\* coacervate leads to further curing of the adhesives and provides strong cohesion to the system.

Dopa-mediated hydrogen bonds are critical to the coacervation of GK-16\*, a peptide derived from the most important mussel adhesive protein, Mfp-5. By forming bidentate hydrogen bonds, Dopa can serve as the "stickers" for the peptide chains, which facilitates the association of the peptides. Such Dopa-mediated interaction can lead to LLPS of the proteins. Additionally, the Gly residues in GK-16\*, which increase the flexibility of the peptide backbone, facilitate the association of the peptides. Our results show that the balance between the electrostatic repulsion and Dopa-mediated hydrogen bonding can regulate the properties of GK-16\*coacervate. Increasing the pH from 3 to 7.4 results in a liquid-to-gel transition of the coacervate.

As GK-16\* resembles the important sequence of Mfp-5, our discoveries can have important biological implications in the formation of the mussel adhesive plaque. Apart from high Dopa content, Mfps share some common features: most Mfps are rich in glycine and all Mfps

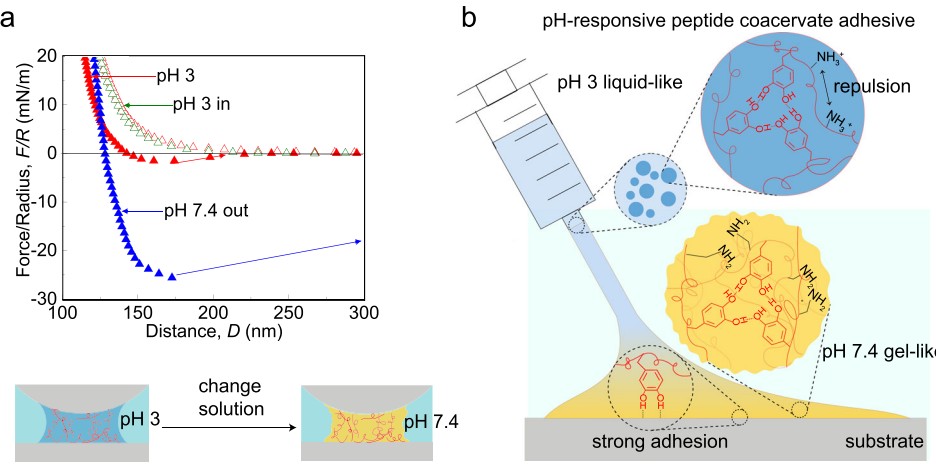

**Fig. 6 | GK-16* based pH-responsive coacervate adhesives. a** SFA measurements on GK-16* coacervate (2 mg/mL, 100 mM ionic strength) indicate a liquid-to-gel transition by changing the solution pH from 3 to 7.4. **b** A schematic of pH-responsive coacervate adhesives. Dopa can serve as the anchoring group for the adhesive on the substrate. The pH-regulated electrostatic repulsion between Lys residues can tune the liquid-to-gel transition of the coacervate adhesive.

have pI above 8, indicating these proteins are positively charged under the seawater environment[38]. As the Mfps are stored and secreted in acidic environment with pH as low as 3[23], the proteins can balance the attractive Dopa-mediated bidentate hydrogen bonds with repulsive electrostatic interactions arising from the positive charges, maintaining the liquid states. After secretion, the high pH (~8) and ionic strength (600 mM) of the sea water can lead to the solidification of the protein matrix. Although GK-16* does not capture all the features of the Mfp-5 protein, it is highly representative of the most important amino acid compositions of Mfp-5, thus the liquid-to-gel transition of the coacervate adhesives may play important roles in the delivery and curing process of natural Mfp-5 during the plaque formation.

In this work, we report that a mussel-inspired peptide, GK-16*, can form coacervate in seawater conditions with a pH-induced liquid-to-gel transition and tunable adhesive properties. By combining point-mutation experiments with MD simulations, our study also demonstrates an effective approach for gaining a deeper understanding of the important interactions that govern the adhesive and phase separation properties of protein and peptide adhesives at the molecular level. With sequence-controlled liquid phase delivery, tunable adhesive and curing properties, as well as excellent biocompatibility, the GK-16*-based peptidyl coacervate adhesives can provide a novel and modular design platform toward the development of functional biomaterials for various applications including wet adhesives and coating materials, tissue glues, drug delivery vehicles, and bioreactors.

## Methods

### Mushroom tyrosinase modification

All variants of GK-16 peptides were purchased from Genscript in which all Y amino acids within sequences were Tyr residues. Mushroom tyrosinase (≥1000 U/mg solid) was purchased from Sigma and all other reagents were of analytical grade. The modification was performed according to previous report[25]. Peptides with the concentration of 1 mg/mL were dissolved in PBS solution (0.1 M, pH 7.4) and borate (0.2 M). Mushroom tyrosinase was then added to initiate the enzymatic modification. Modified peptide solution would turn into a wine color. After 1 h reaction, HCl was added to acidify the solution into pH 3 and filtered with PES (Polyethersulfone) syringe filters of 0.2 μm. The filtered solution would be purified by fast protein liquid chromatography (FPLC) with Superdex 75 HR10/300 GL column using the HCl solvent (1 mM, pH = 3), followed by the dialysis of Mw 1000 under pH 3 HCl solution. After dialysis, GK-16 variant solution was freeze-dried into brown modified power. The middle Tyr residue within three

adjacent Tyr residues in the GK-16 sequence could not be converted into Dopa because of the steric effect[39].

### Circular dichroism (CD)

CD spectra of GK-16 variants were collected on a Chriascan spectropolarimeter (Model 420, AVIV Biomedical Inc.). GK-16 variants were prepared at the concentration of 1 mg/mL in HCl solution (pH 3). Measurements were conducted within wavelengths ranging from 190 to 260 nm, with 1 nm step size and 1 nm bandwidth at 25 °C for three times. The spectra were smoothed by the Savitzky–Golay method with a second-order polynomial. The ionic strength and urea concentration of the solution were tuned by the addition of NaF and urea to maximize the peptide signals.

### Surface force apparatus (SFA) measurements

The adhesion forces between mica surfaces with adsorbed layers of GK-16* peptides and the interfacial energy γ of GK-16* coacervates on mica surfaces were measured using an SFA (SFA 2000) with a reported geometry and procedures[40]. 55 nm silver layer was deposited on the back of mica surfaces which was directly glued on the SFA disks. The distance $D$ between two surfaces was measured with the fringes of equal chromatic order (FECO) technique. 20 μL GK-16* of 2 mg/mL dissolved in HCl solution (1 mM, pH 3) was injected in the gap between two mica surfaces. The system was equilibrated for 30 min before the force measurements.

A series of SFA measurements under pH 3 were conducted to study the adhesive and mechanical properties of the coacervates in solutions of different ionic strength and urea concentrations. Calculated volume of concentrated salt (5 M NaNO₃) or urea (5 M) solution was injected into the gap to change the ionic strength and urea concentration of GK-16* systems to trigger the occurrence of LLPS and/or to change the mechanical properties of the peptide coacervate. To study the adhesive properties of the GK-16* coacervates in different pH values, surfaces were kept in contact with a bridging coacervate film in 1 mM HCl solution (pH 3). The pH of the solution was then changed by replacing the HCl solution with PBS buffer (pH 7.4, 0.1 M). By keeping two surfaces in contact for 30 min, the adhesion forces of the cured GK-16* coacervate was measured during the separation of the surfaces. The normal force $F$ measured in the SFA experiment is normalized by the effective radius of the surfaces, $R$ (typically $R = 2$ cm). The interfacial energy γ conducted from the Young-Laplace equation of the coacervate phase/aqueous solution interface could be simplified as[26,41,42]: $\gamma = F/4\pi R$.

## MD simulations

MD simulations were performed for GK-16 and the mutated variants as well as their Dopa-modified versions based on the GROMACS package[43]. The structure of peptide was predicted using the online server PEP-FOLD3[44]. Following previous studies[45], *Antecamber* module in AMBER[46] was employed to calculate the partial charges and the force field parameters for the Dopa-modified versions. The Amber99ff99SB force field[47] was adopted in the MD simulations. Since the hydroxyl groups of Dopa have a relatively high pKa (pKa1, pKa2 ~ 9.6, 13.1)[48] and our main goal is to investigate the role of Dopa, we adopted the standard protonation states of the amino acids at physiological pH (pH = 7.4) for all peptides. TIP3P water molecules[49] were used to solvate the system; the smooth particle-mesh Ewald method[50] was used to calculate the long-range electrostatic interactions; $Na^+$ and $Cl^-$ ions were added to the system to maintain the specific ionic strength and net charge neutrality. Rigid bonds to hydrogen atoms were constrained using the LINCS algorithm[51]. The simulation box had an initial size around 5 nm, which is large enough to prevent the peptide from interacting with its periodic images. Solvent water molecules and ions were initially relaxed by energy minimization, keeping the peptide atoms restrained. Each system was next subjected to the following protocol: an NVT equilibrium phase for 300 ps where harmonic restraints were placed on the Cα atom with a force constant of 1000 kJ/mol/nm. This was followed by a reduction of the restraints to 100 kJ/mol/nm for 300 ps and then the restraints were reduced to 0 for another 300 ps; For the pair simulations, the cubic system was replicated in the x direction. The equilibrated structures were followed by 100-ns molecular dynamics simulations, and conformations were saved every 10 ps. The time step is 2 fs and the system was modeled as an NPT ensemble (1 atm, 300 K) with periodic boundary conditions in all directions, with temperature and pressure controlled by Nose-Hoover[52] and Parrinello-Rahman[53] algorithms, respectively. The temporal evolutions of the center of mass (COM) distances between the pair of peptides were recorded. The RMSD, RMSF and radius of gyration of the peptides were computed by the GROMACS built-in tools such as *rms*, *rmsf*, *gyrate*, respectively. The trajectories were analyzed using VMD[54] and its extension tools, specifically, the tool *Timeline* for the time evolutions of residue maps of secondary structure and *Hydrogen Bonds* for the time evolutions of the number of hydrogen bonds between the peptides, 0.35 nm and 30 degrees being the cut off distance and angle of hydrogen bonds, respectively. The hydrogen bond occupancy between given residues was taken as the ratio between the total number of hydrogen bonds formed in the trajectories and the total number of trajectory frames, representing the probability of hydrogen bonds in the simulation, the most six probable ones were recorded.

## Optical microscope

Light microscopy images were collected using OLYMPUS IX73 Inverted Microscope. Twenty microliters of GK-16 variant solution was dropped on the glass slides. Figures then visualized at a magnification of ×60. Software Lumenera corporation, Release 6.5, was utilized to analyze images.

## MALDI

AXIMA Performance Shimadzu was used to collect the MALDI spectra of GK-16 variants before and after enzymatic modification. α-cyano-4-hydroxycinnamic acid (CHCA) matrix of 0.7 μL was mixed with peptide solution (2 mg/mL) with a volume ratio of 1:1. All spectra were recorded under the positive mode with the linear MALDI detector.

## UV−Vis

UV−Vis spectra were obtained using UV−Vis-2700 spectrophotometer Shimadzu. UV−Vis cuvettes of 1 mL were selected to record UV−Vis spectra between 200 nm and 800 nm. Calibration was finished before measuring samples.

## Zeta potential

HORIBA SZ-100 Zeta potential analyzer was utilized to measure the zeta potential of GK-16* solution (1 mg/mL) under different pH values based on laser Doppler electrophoresis. Zeta potential glass cell of 100 μL was chosen with measurement angle of 90°. $n$ = 3 biologically independent samples over 3 independent experiments. Data are presented as mean values ± SD.

## Fourier-transform infrared spectroscopy

Fourier-transform infrared spectroscopy experiments were performed with a Bruker Vertex 70 FTIR spectrometer. GK-16* peptides were mixed in 1 mL DCl in $D_2O$ (pD 3) solution with the concentration of 8 mg/mL. Ionic strength was adjusted with the addition of NaCl dissolved in 1 mL DCl in $D_2O$ solution. Peptides were injected into the sealed liquid cell with a pathway of 50 μm. All spectra were measured within the range of 400–4000 $cm^{-1}$ with 1 $cm^{-1}$ resolution and 32 average scans. After obtaining FTIR spectra, OPUS 6.5 software was utilized to analyze the spectra. FTIR spectra of GK-16* in pD 3 solution with various ionic strengths are shown in Supplementary Fig. 3. Due to the concentration difference with the addition of NaCl solution, all spectra are normalized to peak. Their difference spectra clearly show the spectral intensity changes at 1635 and 1668 $cm^{-1}$. The intensities at both peaks rise with increasing ionic strength, implying the correlation between these two vibrational modes. According to their frequencies and intensity ratio, we assign them to the $v_\parallel$ and $v_\perp$ vibrations of β-sheet. The formation of β-sheet with increased ionic strength observed with IR spectra, is consistent with CD results.

## Reporting summary

Further information on research design is available in the Nature Research Reporting Summary linked to this article.

# Data availability

The data generated in this study have been deposited in the DR-NTU(data) under accession code https://doi.org/10.21979/N9/BOOV1A.

# Code availability

Molecular dynamics simulations were performed using GROMACS, which is freely available at https://www.gromacs.org/. Sample input files used to generate the results in this study have been deposited in the DR-NTU(data) under accession code https://doi.org/10.21979/N9/BOOV1A.

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

## Acknowledgements

Q.G., S.C., and J.Y. thank the Singapore National Research Fellowship (NRF-NRFF11-2019-0004) and the Singapore Ministry of Education (MOE) Tier 2 Grant (MOE-T2EP30220-0006). X.Q. and H.G. acknowledge support from the Singapore Ministry of Education (MOE) AcRF Tier 1 (Grants RG138/20). G.Z. and H.G. also acknowledge a start-up grant from Nanyang Technological University and A*STAR, Singapore. The MD simulations reported were performed using resources of the National Supercomputing Centre, Singapore (http://www.nscc.sg).

## Author contributions

Q.G. performed the mussel-derived peptides modification and LLPS characterization experiments and interpreted the results. S.C. assisted within the preparation process. G.Z., X.Q., and H.G. designed, performed, and analysed MD simulations. J.Y. and H.G. directed the studies and wrote the manuscript.

## Competing interests

The authors declare no competing interests.
