## [Peer Review File · Nature Communications]

Hydrogen-bonds mediate liquid-liquid phase separation of mussel derived adhesive peptidesREVIEWER COMMENTS

Reviewer #1 (Remarks to the Author):

This manuscript describes a study aimed to analyze the effect of Dopa residues in peptides to enhance their adhesive ability.

It is well known that mussel-derived adhesive peptides are rich in Dopa residues, so the authors studied a model peptide containing 16 residues, including several Dopa amino acids.

The starting GK-16 contains residues of Gly, Lys, and Tyr. Its treatment with tyrosinase transforms most Tyr residues into Dopa residues, thus leading to the formation of GK-16*. The behavior of the two peptides was analyzed and compared at pH 3 and at 7.4 (in PBS buffer): GK-16* shows a good tendency to form coacervates, while GK-16 does not. Extensive MD simulation was used to analyze these data, including punctual modification of the chains. The general result shows that by increasing the number of Gly residues, the peptide flexibility is increased. Moreover, Dopa is more effective in forming hydrogen bonding than Tyr.

The work is very promising, but the manuscript should be rewritten before publishing in Nature Communications in my opinion. The description of the results is quite confusing and it is not clear which peptides have been prepared and analyzed, or only studied by MD simulation.

Some more work should be done to reach more conclusive results, including:

- Gly was replaced with Ser to check if the peptide flexibility is reduced. Why Ser is used? The authors should explain. Probably Ala would be better, as Ser introduces a hydroxyl group that may form hydrogen bonds, thus giving misleading results.**
- What happens if more Dopa groups are introduced into the structure? And what happens reducing the Lys residues?**
- Is it possible to choose the best candidate for adhesion, make it, and check if the expected results are obtained?**

Reviewer #2 (Remarks to the Author):

Using a combination of experimental coacervation studies on synthetic peptides and MD computational analysis, the authors put forth an interesting hypothesis that hydrogen bonding involving DOPA and possibly glycine can initiate LLPS of a short 16 amino acid peptide based on a sequence found in the mussel adhesive proteins mfp-5. This hypothesis was supported by a systematic study of the behavior of the peptide following a series of targeted point mutations in the sequence. The data are very interesting and compelling, although there are several points that the authors need to address.

1. In the introduction, the authors state "In the mussel foot, the Mfps form coacervate triggered by the sea water chemical environment...". Although the plaque must function in seawater, I don't think it is clear that coacervation is triggered by seawater. Based on the work of Waite, it appears that the mussel controls the internal environment of the foot distal depression during secretion (Waite 2017 J. Exp. Biol.). Probably seawater plays an important role in plaque maturation and cross-linking over time based on the recent work of Valentine and Waite (Bernstein 2020 Soft Matter) rather than initial LLPS. The authors should adjust their statement accordingly.

2. While the GK-16 peptide does have many features that are relevant to mfp-5 and other byssus adhesive and coating proteins (i.e. Gly, Lys and Tyr/DOPA), it is not entirely representative of the whole mfp-5 sequence. As clear from figure S1, there are significant regions of the protein that are very different from this sequence. In particular, the N-terminal region contains quite a few hydrophobic valine residues, as well as negatively charged aspartate residues and pH-responsive Histidine residues (which will be positively charged at acidic pH). In light of these large differences, it seems a stretch based on the current data to identify the GK-16 as the source of the

phase separation behavior in the native protein. I agree that it possibly plays a role in the natural system, but it is premature to insist on this point now. The authors should clarify this point in their paper.

3. I'm curious why the authors decided to primarily use sodium nitrate rather than sodium chloride for their coacervation experiments. What was the motivation here? If the goal is to mimic seawater, then there would be a much higher concentration of chloride anions than nitrate. The authors mentioned trying NaCl but did not mention if there were any differences in the coacervation behavior compared with NaNO₃. Can the authors please clarify this?

4. I'm curious how the CD spectroscopy data were analyzed to determine the beta sheet content of the solution. I can see changes in the spectra but based on my understanding they do not appear to show a very strong beta sheet signal. Were the data fitted? Did the authors consider attempting other methods to corroborate these conformational assignments like FTIR or Raman spectroscopy? I found this to be a weak point of the manuscript. The hypothesis would be significantly enhanced with addition of vibrational spectroscopy data.

5. Considering that pH is such a critical aspect of coacervation and formation of the byssus adhesion, I was wondering how pH was handled in the MD simulations. This will have an effect on the protonation state of the DOPA residues, which could influence their tendency to hydrogen bond. Can the authors please clarify?

6. I found it very interesting that replacing DOPA with glycine did not have a massive change on the LLPS behavior. Glycine residues in silkworm silk (GA) repeat sequences also form H-bonds through backbone carbonyls and amide groups surrounding the peptide bond. These beta sheets are then further stabilized in the native silk via beta sheet stacking and formation of crystallites enabled via hydrophobic interactions between the Gly and Ala side chains on opposite sides of the beta sheet. In this case, beta sheet formation does not require DOPA at all and tyrosine tends to disrupt crystallite stacking. My question is whether the authors are somehow converting the GK-16 sequence from mfp-5-like to a silk-like sequence (also prone to coacervation) through their point mutations. This is an interesting, albeit more philosophical, point to consider.

7. Related to the last point, if the backbone carbonyls and amide groups are what is responsible for forming beta sheet structure (as suggested by the authors), then the DOPA side chains would be facing into the solution off the face of the sheet, free to interact with other beta sheets, which could lead to formation of crystallites. This could be proven with X-ray diffraction or at least further supported with IR/Raman. Could the authors please comment on this?

8. I found the SFA adhesion data in Figure 6 perplexing. These pH dependent data are almost the complete opposite of what was previously published in Yu et al 2011 Nature Chemical Biology in which the strongest adhesion for mfp-3 was observed at low pH with reduced adhesion at higher pH. Can the authors offer an explanation for this behavior? Mfp-3 and mfp-5 are somewhat similar in terms of sequence, so I would not expect the SFA adhesion behavior to be so different.

Response to Reviewers

Reviewer #1

Summary comments:

This manuscript describes a study aimed to analyze the effect of Dopa residues in peptides to enhance their adhesive ability. It is well known that mussel-derived adhesive peptides are rich in Dopa residues, so the authors studied a model peptide containing 16 residues, including several Dopa amino acids. The starting GK-16 contains residues of Gly, Lys, and Tyr. Its treatment with tyrosinase transforms most Tyr residues into Dopa residues, thus leading to the formation of GK-16*. The behavior of the two peptides was analyzed and compared at pH 3 and at 7.4 (in PBS buffer): GK-16* shows a good tendency to form coacervates, while GK-16 does not. Extensive MD simulation was used to analyze these data, including punctual modification of the chains. The general result shows that by increasing the number of Gly residues, the peptide flexibility is increased. Moreover, Dopa is more effective in forming hydrogen bonding than Tyr.

The work is very promising, but the manuscript should be rewritten before publishing in Nature Communications in my opinion. The description of the results is quite confusing and it is not clear which peptides have been prepared and analyzed, or only studied by MD simulation.

Our response:

We are grateful for the reviewer's valuable comments. We have performed additional tests according to the suggestion of the reviewer and have revised our manuscript to incorporate the new results. We would like to clarify that all the peptides have been prepared and analysed by both experiments and MD simulations. This is shown in Figure 4A, which shows a very good agreement between the results from phase separation experiments and associated results from MD simulation.

Some more work should be done to reach more conclusive results, including:
- Gly was replaced with Ser to check if the peptide flexibility is reduced. Why Ser is used? The authors should explain. Probably Ala would be better, as Ser introduces a hydroxyl group that may form hydrogen bonds, thus giving misleading results.

Our response:

We appreciate reviewer's comments. Within native Mfp-5 protein, Ser amino acid takes a percentage of 8.3% (Zhao H, Waite J H. *Journal of Biological Chemistry*, 2006, 281(36): 26150-26158.). Gly and Ser are the commonly used flexible spacers for peptide design (Argos P. *Journal of molecular biology*, 1990, 211(4): 943-958.). Replacing Gly into Ser would affect the flexibility of the peptide chain. We agree with the reviewer that if the hydroxyl group of Ser can form extra hydrogen bonds and then facilitate the association, it will give misleading results. However, our MD simulations and experiments showed the opposite, i.e., the total hydrogen bond numbers of GK-16-2S3S* is less than that of GK-16-2G3G* (Fig. S23). Replacing Gly into Ser mainly changes the interaction pair points from main-main to side-main (Fig. 5B (iv) and Fig.S23). Completely changing Gly to Ser leads to no association in simulation or LLPS in experiments albeit the existence of the hydroxyl groups (Fig. 4 and S19). Above all, instead of giving misleading results, our findings further strengthen our conclusions that flexibility is very important.

Introducing Ala into the sequence would bring hydrophobic interactions into consideration. (Lefèvre T, Rousseau M E, Pézolet M. *Biophysical journal*, 2007, 92(8): 2885-2895.). Although by replacing Gly with Ser/Ala, the flexibility of the chain will be reduced (Fig. R1A), for the case with Ala, the hydrophobic interaction between peptides should be enhanced. Here, we designed point-mutated peptide: GK-16-2A3A (Table R1 and Fig. R2). The tyrosine residue is modified into Dopa and we compare its phase separation behavior with GK-16-2G3G* and GK-16-2S3S*.

Figure R1. MD simulations of GK-16-2A3A and its Dopa-modified conversion: (A) The root-mean-square fluctuations (RMSFs) of original and Dopa modified GK-16, GK-16-2G3G, GK-16-2S3S, GK-16-2A3A; (B) The representative temporal evolutions of COM distances between the pairs of GK-16-2A3A, GK-16-2A3A*; (C) Quantification of the number of H-bonds formed as a function of time during the 100 ns MD simulation; (D) The occupancy of H-bond between different residues was obtained from the last 40 ns of simulations of GK-16-2A3A*; (E) The representative configurations of GK-16-2A3A and its modification in MD simulations.

Table R1. Molecular mass before and after enzymatic modification, predicted pI value, abbreviation and sequence of GK-16-2A3A.

Abbreviation	Sequence	Mw	pI	Whether LLPS	
				Unmodified	Modified
GK-16	GYKGYGKGGKYYK	1994	9.85	×	√
GK-16-2A3A	GYKGAAGKGGKAAK	1534	10.4	×	√

Figure R2. (A) MALDI spectra before and after enzymatic modification of GK-16-2A3A. (B) Optical microscopic images of GK-2A3A* (2 mg/mL) coacervates. (C) CD spectra of GK-16-2A3A* (1 mg/mL) with different ionic strengths. (D) SFA force-distance profiles of GK-16-2A3A* (2 mg/mL) in solutions with different ionic strengths.

By replacing Gly with Ala, GK-16-2A3A* showed a larger phase separation region than GK-16-2S3S*, but a smaller LLPS region than GK-16-2G3G* (Fig R3). The phase separation results support our hypothesis that Ala increases the hydrophobic interactions between peptide chains, while decreases the chain flexibility. Consistent with the experiments, our MD simulations also showed a preferred association for GK-16-2A3A* compared to GK-16-2A3A (Fig. R1B and F) with most hydrogen bonds formed between backbones likely due to the hydrophobic interactions (Fig. R1C and D). CD spectra of GK-16-2A3A* indicates that introducing Ala into the sequence would promote the formation of α -helix (Fig. R2C). GK-16-2A3A* formed coacervate droplets with smaller radii (Fig. R2B) under the microscope. SFA measurements confirmed that GK-16-2A3A* is more gel-like with the increasing ionic strength. (Fig. R2D).

Figure R3. Phase diagrams of GK-16* variants: (A) GK-16-2G3G*; (B) GK-16-2S3S*; (C) GK-16-2A3A*.

Modifications to the manuscript:

1. Page 9. We have added the sentence. “In addition, we replaced the Gly in GK-16-2G3G* with less flexible Ala or Ser residues^{30, 31}. With such mutations, GK-16-2S3S* showed a much narrower LLPS region than GK-16-2G3G* (Fig. 4B), while GK-16-2A3A* showed a larger phase separation region than GK-16-2S3S*, but a smaller LLPS region than GK-16-2G3G* (Fig. 4B). The differences in the phase separation regimes of GK-16-2S3S* and GK-16-2A3A* is likely due to the stronger hydrophobicity of Ala. ”

2. Page 12. We have added. “There are two expected competing roles for the extra hydroxyl side group of Ser residue: on one side, it increases the hydrogen bond capacity while on the other side, it can also make the peptide less conformable (Fig. S18)³⁰. Our simulations suggest that the latter effect will be more important as GK-16-2S3S* exhibits a relative loose association compared to GK-16-2G3G* (Fig. S19 and S21), with slightly less H-bonds.... Mutating Gly by Ala at position-2 and -3 (GK-16-2A3A*) will help the backbone hydrogen bond formations and association (Fig. 5A (x), S19, S21I, S23) likely due to the hydrophobic interactions.”

3. We have incorporated Fig. R1A into Fig. S18; Fig.R1B into Fig. S19; Fig.R1C into Fig. S23; Fig. R1D into Fig. 5A(x); Fig.R1E into Fig. S20 and S21.

4. We have incorporated Table R1 into Table S2 and Fig. 4A.

5. We have incorporated Fig. R2A into Fig. S5K; Fig. R2B into Fig. S12J; Fig. R2C into Fig. S14J; Fig. R2D into Fig. S26G.

6. We have incorporated Fig. R3C into Fig. 4B(x).

- What happens if more Dopa groups are introduced into the structure? And what happens reducing the Lys residues?

Our response:

Thanks for the suggestion. We designed GK-16-GG/YY to check the influence of more Tyr residues by replacing two Gly (G1 and G4) into Tyr residues (Table R2). We also designed GK-16-KK/GG and GK-16-K/G to check the influence of reducing Lys residues. GK-16-KK/GG replaced two Lys (K11 and K12) into Gly residues (Table R2). We also replaced all Lys into Gly of GK-16. However, the peptide GK-16-K/G has a poor solubility, which could not be enzymatically modified into Dopa version.

Table R2. The list of GK-16, GK-16-GG/YY, GK-16-KK/GG and GK-16-K/G with modified sequences, and corresponding LLPS and association behaviors in experiments.

Abbreviation	Sequence	Mw	pI	Whether LLPS	
				Unmodified	Modified
GK-16	GYK GK YY GK GK K YY YK	1994	9.85	×	√
GK-16-GG/YY	YY K YK YY GK GK K YY YK	2207	9.75	×	√
GK-16-KK/GG	GYK GK YY GK GGG YY YK	1582	9.63	×	√
GK-16-K/G	GY GGG YY GGG GG YY YG	1568	5.52	N/A	N/A

Figure R4. MALDI spectra before and after enzymatic modification of (A) GK-16-GG/YY; (B) GK-16-KK/GG.

For GK-16-GG/YY*, the main MALDI peak indicates that only 5 Tyr residues are successfully modified into Dopa residues because of the steric effect (Fig. R4A). GK-16-GG/YY* and GK-16-KK/GG could both undergo LLPS (Fig. R5). GK-16-GG/YY* showed a larger phase separation region than that of GK-16*; while GK-16-KK/GG* showed a slightly larger LLPS region compared with GK-16* (Fig. R6).

Figure R5. Optical microscopic images of GK-16 variant (2 mg/mL) coacervates. (A) GK-16-GG/YY*; (B) GK-16-KK/GG*.

Figure R6. Phase diagrams of GK-16* variants: (A) GK-16*; (B) GK-16-GG/YY*; (C) GK-16-KK/GG*.

Figure R7. MD simulations of pair interactions of GK-16-GG/YY, GK-16-KK/GG and the Dopa-modified versions. (A) The representative temporal evolutions of COM distances between the pairs of GK-16-GG/YY, GK-16-GG/YY*; (B) Quantification of the number of H-bonds formed as a function of time during the 100 ns MD simulation; (C) The occupancy of H-bond between different residues was obtained from the last 40 ns of simulations of GK-16-GG/YY*; (D) The representative configurations of GK-16-GG/YY and GK-16-GG/YY* in MD simulations; (E) The representative temporal evolutions of COM distances between the pairs of GK-16-KK/GG, GK-16-KK/GG*; (F) Quantification of the number of H-bonds formed as a function of time during the 100 ns MD simulation; (G) The occupancy of H-bond between different residues was obtained from the last 40 ns of simulations of GK-16-KK/GG*; (H) The representative configurations of GK-16-KK/GG and GK-16-KK/GG* in MD simulations.

We also performed the MD simulations to investigate the pair interactions and associations of GK-16-YY/GG, GK-16-KK/GG and their derivatives (Fig. R7). Our simulations showed that introducing more Dopa residues increases the hydrogen bonds between peptide chains (Fig. R7B,C and Fig. S23), and therefore leading to a larger LLPS regime in experiments (Fig.

R6). Meanwhile, reducing the number of Lys residues in the sequence reduces the electrostatic repulsion between peptides, which could also promote the association among peptides. It is confirmed by simulations that GK-KK/GG* can associate under a relatively low ionic strength (Fig.R7E and R7H). CD spectra revealed that both GK-16-GG/YY* and GK-16-KK/GG* would form β -sheets with the increasing ionic strength (Fig. R8). Notably, by replacing Gly into Dopa residues, GK-16-GG/YY* also showed viscous jump out in the SFA test, indicating that the GK-16-GG/YY* coacervate remained liquid-like under 600 mM ionic strength as shown in Fig. R9.

Figure R8. CD spectra of GK-16 derived peptides (1 mg/mL). (A) GK-16-GG/YY*; (B) GK-16-KK/GG*.

Figure R9. SFA force-distance profiles of GK-16-GG/YY* (2 mg/mL) in solutions with different ionic strengths.

Modifications to the manuscript:

1. Page 9. We have added the sentence. “By replacing two Gly residues of GK-16* into Dopa, GK-16-GG/YY* showed a slightly larger LLPS region (Fig. 4B) as increasing the amount of Dopa residues could promote the hydrogen bonds.”

2. Page 10. We also have added the sentence. “We also explored the effect of Lys in the LLPS behavior of GK-16*. Completely replacing the positively charged Lys to Gly rendered the peptide insoluble in aqueous solution. We then replaced 2 Lys residues of GK-16* with Gly (GK-16-KK/GG*) to reduce the charge of the peptide. GK-16-KK/GG* showed a slightly larger phase separation region than that of GK-16* because of the reduced electrostatic repulsion (Fig. S15), which is also supported by the association of GK-16-KK/GG* under lower ionic strength in simulations (Fig. 4A, S19, S21K). Such point mutations demonstrate the importance of positive charges in balancing the solubility and phase separation behavior of the peptide.”

3. Page 11. We also have added the sentence. “With increased content of Dopa, more hydrogen bonds were observed between Dopa and other residues (Fig. 5A(ii), S23).”

4. We have incorporated Table R2 into Table S2 and Fig. 4A.

5. We have incorporated Fig. R4A and R4B into Fig. S5B and S5M, separately.

6. We have incorporated Fig. R5A and R5B into Fig. S12B and S12K, separately.

7. We have incorporated Fig. R6B into Fig. 4B(ii); Fig. R6C into Fig. S15.

8. We have incorporated Fig. R7A into Fig. S19; Fig.R7B into Fig. S23; Fig.R7C into Fig. 5A(ii); Fig. R7D into Fig. S20 and S21; Fig. R7E into Fig. S19; Fig.R7F into Fig. S23; Fig.R7G into Fig. S22B; Fig. R7H into Fig. S20 and S21.

9. We have incorporated Fig. R8A and R8B into Fig. S14A and S14L, separately.

10. We have incorporated Fig. R9 into Fig. S26A.

- Is it possible to choose the best candidate for adhesion, make it, and check if the expected results are obtained?

Our response:

We systematically tested the adhesive properties of GK-16* and its variants, and the results are the summary Table R3. Under 1 mM ionic strength, all peptides show adhesion because of the electrostatic attraction between peptide layers and mica surfaces (Fig. R10). By adding ionic strength to 100 mM, they all show phase separation behaviours which are indicated by the increasing film thickness. Under 100 mM ionic strength, GK-16*, GK-16-GG/YY*, GK-16-2G* and GK-16-2G3G* and all exhibit viscous jump out, which indicates these peptides

form liquid-like coacervates (Fig. 2D, Fig. R10A, C and F). On the contrary, GK-16-1G*, GK-16-3G*, GK-16-1G2G* and GK-16-2A3A* form more gel-like complexes (Fig. R10B, D, E and G). By further increasing the salt concentration to 600 mM, only GK-16*, GK-16-GG/YY* and GK-16-2G* show viscous jump out with bridging SFA profiles (Fig. 2D, Fig. R11A and C).

Table R3. Summary of adhesion forces (mN/m) measured by SFA for different GK-16 derived peptides (2 mg/mL) with different ionic strength.

Abbreviation	1 mM	100 mM	600 mM	2000 mM
GK-16*	-1.05±0.08	-1.24±0.13	-1.34±0.14	N/A
GK-16-GG/YY*	-1.73±0.24	-1.60±0.41	-0.74±0.06	N/A
GK-16-1G*	-1.22±0.08	N/A	N/A	N/A
GK-16-2G*	-2.92±0.58	-0.66±0.09	-0.85±0.25	N/A
GK-16-3G*	-2.68±0.24	N/A	N/A	N/A
GK-16-1G2G*	-0.64±0.09	N/A	N/A	N/A
GK-16-2G3G*	-1.84±0.28	-0.59±0.06	N/A	N/A
GK-16-2A3A*	-0.72±0.14	N/A	N/A	N/A

Figure R10. SFA force-distance profiles of different GK-16 derived peptides (2 mg/mL) in solutions with different ionic strengths. (A) GK-16-GG/YY*; (B) GK-16-1G*; (C) GK-16-2G*; (D) GK-16-3G*; (E) GK-16-1G2G*; (F) GK-16-2G3G*; (G) GK-16-2A3A*.

We then tested the pH-dependent adhesive properties for GK-16*, GK-16-GG/YY* and GK-16-2G*. For GK-16-GG/YY*, the SFA profile indicates the complexes are more viscous than those of GK-16* with the same 4 cycle decay period (Fig. R11 A, B). However, the adhesion force of GK-16-GG/YY* is only half value of that of GK-16*. For GK-16-2G*, after changing pH from 3 to 7.4, GK-16-2G* also shows strong cohesion failure with adhesion forces decayed to zero after 4 cycles (Fig. R11D). However, compared to GK-16* and GK-16-GG/YY*, GK-16-2G* exhibits more rigid complexes without a viscous bridging jump out (Fig. R11C). The adhesion force of GK-16-2G* is also weaker than GK-16*. On the summary, GK-16* is the best candidate for pH-induced coacervate adhesives.

Figure R11. SFA measurements on (A) GK-16-GG/YY* coacervate and (C) GK-16-2G* (2 mg/mL, 100 mM ionic strength) indicate a liquid-to-gel transition by changing the solution pH from 3 to 7.4. SFA force-distance profiles of (B) GK-16-GG/YY* and (D) GK-16-2G* (2 mg/mL) in PBS buffer (0.1 M, pH = 7.4) showed that the adhesion force rapidly decreased to zero after 4 cycles.

Modifications to the manuscript:

1. Page 13. We have added the sentence. “It is worth mentioning that such pH responsive adhesiveness would require balanced molecular interactions between the peptides. Too strong interactions would lead to gel-like coacervates at high salt concentrations even at pH 3, which abolished the adhesion of the peptide coacervates (Table S3, Fig S26). Apart from GK-16*, GK-16-GG/YY* and GK-16-2G* also demonstrated such pH responsive liquid-to-gel transition (Fig S27).”
2. We have added Fig. 10 as Fig. S26 into the supporting information.
3. We have added Fig. R11 as Fig. S27 into the supporting information.
4. We have added Table R3 as Table S3 into the supporting information.

Reviewer #2

Summary comments:

Using a combination of experimental coacervation studies on synthetic peptides and MD computational analysis, the authors put forth an interesting hypothesis that hydrogen bonding involving DOPA and possibly glycine can initiate LLPS of a short 16 amino acid peptide based on a sequence found in the mussel adhesive proteins mfp-5. This hypothesis was supported by a systematic study of the behavior of the peptide following a series of targeted point mutations in the sequence. The data are very interesting and compelling, although there are several points that the authors need to address.

Our response:

We are grateful for the reviewer’s valuable comments.

1. In the introduction, the authors state “In the mussel foot, the Mfps form coacervate triggered by the sea water chemical environment...”. Although the plaque must function in seawater, I don’t think it is clear that coacervation is triggered by seawater. Based on the work of Waite, it appears that the mussel controls the internal environment of the foot distal depression during secretion (Waite 2017 J. Exp. Biol.). Probably seawater plays an important role in plaque maturation and cross-linking over time based on the recent work of Valentine and Waite (Bernstein 2020 Soft Matter) rather than initial LLPS. The authors should adjust their statement accordingly.

Our response:

We have revised our manuscript on Page 2 as “In the mussel foot, Mfps form coacervates in the internal environment of the foot distal depression during secretion. After spatial-temporally

regulated¹⁰ secretion of Mfps, the seawater triggers the solidification of the secreted coacervates and further plaque maturation¹¹.”

2. While the GK-16 peptide does have many features that are relevant to mfp-5 and other byssus adhesive and coating proteins (i.e. Gly, Lys and Tyr/DOPA), it is not entirely representative of the whole mfp-5 sequence. As clear from figure S1, there are significant regions of the protein that are very different from this sequence. In particular, the N-terminal region contains quite a few hydrophobic valine residues, as well as negatively charged aspartate residues and pH-responsive Histidine residues (which will be positively charged at acidic pH). In light of these large differences, it seems a stretch based on the current data to identify the GK-16 as the source of the phase separation behavior in the native protein. I agree that it possibly plays a role in the natural system, but it is premature to insist on this point now. The authors should clarify this point in their paper.

Our response:

We would like to thank the review for this comment. Although our peptide does not capture all the features of the whole sequence of Mfp-5 as the reviewer pointed out, it does represent the key amino acid composition of Mfp-5. We have changed our expression in the conclusion session on Page 17 to “Although GK-16 does not capture all the features of the Mfp-5 protein, it is highly representative of the most important amino acid compositions of Mfp-5, and the liquid-to-gel transition of the coacervate adhesives may play important roles in the delivery and curing process of natural Mfp-5 during the plaque formation.”

3. I’m curious why the authors decided to primarily use sodium nitrate rather than sodium chloride for their coacervation experiments. What was the motivation here? If the goal is to mimic seawater, then there would be a much higher concentration of chloride anions than nitrate. The authors mentioned trying NaCl but did not mention if there were any differences in the coacervation behaviour compared with NaNO₃. Can the authors please clarify this?

Our response:

We chose NaNO₃ in our experiments as it is more suitable for SFA experiment. SFA experiments uses mica surfaces evaporated with silver on the back side of the mica. The silver layer functions as the mirror in the SFA FECO technique. High concentration of chlorine can cause the oxidation of the back silver layer, which can lead to the failure of the SFA experiment. The type of salt may influence the phase separation of some coacervation systems (Zhang Y, Cremer P S. *Current opinion in chemical biology*, 2006, 10(6): 658-663; Perry S L, Li Y, Priftis D, et al. *Polymers*, 2014, 6(6): 1756-1772.), and we have tested the phase behavior of GK 16*

using NaCl and NaNO₃, and no obvious difference was observed (Fig. R12). Therefore, we chose NaNO₃ instead of NaCl.

Figure R12. Phase diagram of GK-16* with different anions. (A) NaCl; (B) NaNO₃.

Modifications to the manuscript:

1. Page 4. We have edited the sentence. “LLPS occurred when GK-16* concentration was beyond 0.5 mg/mL and NaNO₃ or NaCl concentration was above 100 mM (Fig. S2).”

2. We have added Fig. R12 as Fig. S2 into the supporting information.

4. I’m curious how the CD spectroscopy data were analyzed to determine the beta sheet content of the solution. I can see changes in the spectra but based on my understanding they do not appear to show a very strong beta sheet signal. Were the data fitted? Did the authors consider attempting other methods to corroborate these conformational assignments like FTIR or Raman spectroscopy? I found this to be a weak point of the manuscript. The hypothesis would be significantly enhanced with addition of vibrational spectroscopy data.

Our response:

Thank for the advice. Using Bestsel software, CD data have been fitted. Below are the estimated secondary structure contents of CD spectra.

Table R4. Fitting CD data of GK-16* with Bestsel software with ionic strength of 1 mM, 50 mM and 600 mM.

Ionic strength (mM)	α -helix (%)	β -sheet (%)	β -turn (%)	Random coil (%)
1	23.8	0	0	76.2
50	6.4	36.4	14.3	42.8
600	0	53.9	6.3	39.8

The estimated secondary structure contents indicate the increasing content of β -sheet composition with the increasing ionic strength.

FTIR spectra of GK-16* in pH 3 solution with various ionic strength are shown in Figure R13. Due to the concentration difference with the addition of NaCl solution, all spectra are normalized to peak. Their difference spectra clearly show the spectral intensity changes at 1635 and 1668 cm^{-1} . The intensities at both peaks rise with increasing ionic strength, implying the correlation between these two vibrational modes. According to their frequencies and intensity ratio, we assign them to the ν_{\parallel} and ν_{\perp} vibrations of β -sheet. The formation of β -sheet with increased ionic strength observed with IR spectra, is consistent with our CD results.

Figure R13. FTIR spectra of GK-16* (5 mg/mL) in pH 3 DCl solution with ionic strength of 1 mM, 600 mM and 2000 mM and their difference spectra compared to the one measured with ionic strength of 1mM. All spectra are normalized to peak.

Modifications to the manuscript:

1. Page 3. We have modified the sentence. “CD and IR spectra reveals that the second structure of the peptide changes from mainly random-coil to a mixture of coil and β -sheet structures as the ionic strength of the solution increased from 1 mM to 600 mM (Fig. 2C, Fig. S3, and Table S1).”

2 We have included IR measurements and discussion of the IR results into the methods session of the supporting information.

3. We have added Table R4 as Table S1 into the supporting information.

4. We have added Fig. R13 as Fig. S3 into the supporting information.

5. Considering that pH is such a critical aspect of coacervation and formation of the byssus adhesion, I was wondering how pH was handled in the MD simulations. This will have an effect on the protonation state of the DOPA residues, which could influence their tendency to hydrogen bond. Can the authors please clarify?

We agree with the reviewer that pH is very important in the adhesion properties of the coacervates (Fig. 6). However, in terms of the coacervation capability, Dopa modification is critical while pH only shifts the LLPS boundaries (Fig. 2B). Our MD simulations were designed to help investigate the effect of Dopa residues, so we adopted the standard protonation states of the amino acids at physiological pH (pH=7.4) for all peptides. For Dopa, the pKa1 of the Dopa residues is around 9.6 (Charkoudian, L.K. et al., *Inorg. Chem.* 2006, 45, 3657–3664), based on the Henderson-Hasselbach equation ($\text{pH} = \text{pKa} + \log_{10}[\text{O}^-]/[\text{OH}]$), at 7.4, the deprotonated portion of the hydroxyl group $[\text{O}^-]/[\text{OH}]$ is around $10^{-2.2}$, which is almost negligible (<1%). At acidic condition (pH=3), the deprotonated portion will be reduced to around $10^{-6.6}$. These estimations indicate that the deprotonated Dopa wouldn't play a significant role in the association process. In addition, directly modelling the pH in MD simulation is challenging and computationally very expensive (Radak, BK. et al., *J. Chem. Theory Comput.* 2017, 13 (12): 5933-5944.), which makes it impractical for our systems. Above all, we believe our MD simulations reasonably captured the protonation state of Dopa under current modelling and computational capabilities. We have included these clarifications in our manuscript.

Modifications to the manuscript:

1. Page 17. We included the clarifications: “Since the hydroxyl groups of Dopa have a relatively high pKa (pKa1, pKa2 ~ 9.6, 13.1)⁵⁰ and our main goal is to investigate the role of Dopa, we adopted the standard protonation states of the amino acids at physiological pH (pH=7.4) for all peptides.”

6. I found it very interesting that replacing DOPA with glycine did not have a massive change on the LLPS behavior. Glycine residues in silkworm silk (GA) repeat sequences also form H-bonds through backbone carbonyls and amide groups surrounding the peptide bond. These beta sheets are then further stabilized in the native silk via beta sheet stacking and formation of crystallites enabled via hydrophobic interactions between the Gly and Ala side chains on opposite sides of the beta sheet. In this case, beta sheet formation does not require DOPA at all and tyrosine tends to disrupt crystallite stacking. My question is whether the authors are somehow converting the GK-16 sequence from mfp-5-like to a silk-like sequence (also prone to coacervation) through their point mutations. This is an interesting, albeit more philosophical, point to consider.

Our response:

Thank for the comments. We did not attempt to convert the sequence to a silk-like sequence in our mutation experiments. Silk fibroin proteins are mainly composed of Ala (4-57 mol%), Ser (0-40 mol%), Tyr (5-10 mol%), and Gly (2-48 mol%) varied by species (Sutherland T D, Young J H, Weisman S, et al. *Annual review of entomology*, 2010, 55: 171-188.). Although replacing Dopa with Gly in the sequence does make the sequence more silk-like, the sequence of Mfp-5 and GK 16 are very different from that of silk. Native Mfp-5 contains low content of Ala, which is abundant in the silk. With the mutations we performed, GK-16* variants become more silk-like with high Gly and Tyr contents. However, our peptides are still highly charged due to the high Lys content, whereas there are much fewer charged residues in the silk.

7. Related to the last point, if the backbone carbonyls and amide groups are what is responsible for forming beta sheet structure (as suggested by the authors), then the DOPA side chains would be facing into the solution off the face of the sheet, free to interact with other beta sheets, which could lead to formation of crystallites. This could be proven with X-ray diffraction or at least further supported with IR/Raman. Could the authors please comment on this?

Our response:

We could not perform XRD test on the liquid coacervate samples. As CD spectra and FTIR spectra indicate that our GK-16* peptides show an increasing content of β -sheet with the increasing ionic strength (Fig. R13 and Table R4). However, GK-16* cannot form regular β -sheet structures in the liquid-status coacervate droplets. We have commented on the increasing of beta sheet β -sheet with the increasing ionic strength on page 4.

8. I found the SFA adhesion data in Figure 6 perplexing. These pH dependent data are almost the complete opposite of what was previously published in Yu et al 2011 Nature Chemical Biology in which the strongest adhesion for mfp-3 was observed at low pH with reduced adhesion at higher pH. Can the authors offer an explanation for this behavior? Mfp-3 and mfp-5 are somewhat similar in terms of sequence, so I would not expect the SFA adhesion behavior to be so different.

Our response:

Thank for the comments. There is no conflict between our coacervate adhesion measurements and previous SFA measurements on the adhesive properties of Mfps. In the 2011 Nature Chemical Biology paper by Yu et al, SFA was used to measure the adhesion of Mfp-3 monolayer adsorbed on the surface. The adhesion reported was mainly due to the adhesion between the Mfp-3 layer and mica surfaces. The adhesion of Mfp-3 was diminished after changing pH from 3 to 7. This is due to the oxidation of Dopa into Dopakinone, which renders Dopa's ability of forming bidentate hydrogen bonds with mica surfaces. However, in our current SFA experiments, we measured the adhesion of GK-16* coacervate. At pH 3, the adhesion measured was due to the interfacial tension of the GK-16* coacervate. Similar to many coacervate systems (Wei W, Tan Y, Rodriguez N R M, et al. Acta biomaterialia, 2014, 10(4): 1663-1670; Hwang D S, Zeng H, Srivastava A, et al. Soft matter, 2010, 6(14): 3232-3236.), the GK-16* coacervate showed low interfacial tension, and therefore the adhesion force measured was small. The liquid state of the GK-16* coacervate is evident by the very reproducible adhesion force measured by the SFA in repeating force runs (Fig. S24A), showing that the coacervate layer coated on both mica surfaces can coalesce into one liquid film. When changing the buffer pH to 7, we kept two surfaces in contact. So the two surfaces were 'glued' together by the GK-16* coacervate. At pH 7, the coacervate become more gel-like, and therefore the adhesion force measured by the SFA is the force associated with the cohesive failure of the gel. Therefore, the adhesion measured at pH 7 is 15-time stronger than that of pH 3. As the GK-16 coacervate gel was gel like, after the cohesive failure of the gel in the initial force run, the two gel-like layers cannot coalesce into one film. Due to such cohesive failure

of the GK-16 gel, followed by molecular rearrangement and Dopa oxidation, the adhesion force measured at pH 7 rapidly decreased in the following force runs.

Figure R14. Illustration of pH-dependent SFA experiments. (A) Peptides form liquid-like condensates at pH3; (B) After keeping surfaces in contact and changing pH to 7, complexes become more gel-like. (C) Upon separation, the cohesive failure happens with a large adhesion force.

Modifications to the manuscript:

1. Page 13. We have modified the text as “Such strong adhesion was only measured in the first separation, and the adhesion forces measured in the subsequent force runs rapidly decreased to zero (Fig. S24B and S25). This is due to the cohesive failure of the gel-like GK-16* coacervate adhesive at pH 7.4 during the first separation of two surfaces.”

2. We have added Fig. R14 as Fig. S25 into the supporting information.

REVIEWERS' COMMENTS

Reviewer #1 (Remarks to the Author):

The manuscript may be published in the current form.

Reviewer #2 (Remarks to the Author):

The authors have sufficiently addressed my previous concerns. It's a very nice paper.